

# Development and internal and external validation of a nomogram model for predicting the risk of chronic kidney disease progression in IgA nephropathy patients

Ying Zhang[1], Zhixin Wang[1], Wenwu Tang[2], Xinzhu Yuan[1] and Xisheng Xie[1,3,4]

[1] Department of Nephrology, Nanchong Central Hospital Affiliated to North Sichuan Medical College, Nanchong, Sichuan, China
[2] Department of Nephrology, Guangyuan Central Hospital, Guangyuan, Sichuan, China
[3] Nanchong Key Laboratory of Basic and Clinical Research of Chronic Kidney Disease, Nanchong, Sichuan, China
[4] Nanchong Clinical Medical Research Center, Nanchong, Sichuan, China

Corresponding author
Xisheng Xie,
xishengxie2023@163.com

## ABSTRACT

**Background**. IgA nephropathy (IgAN) is the most common primary glomerular disease in chronic kidney disease (CKD), exhibiting significant heterogeneity in both clinical and pathological presentations. We aimed to explore the risk factors influencing short-term prognosis ($\geq$90 days) and to construct a nomogram model for evaluating the risk of CKD progression in IgAN patients.

**Methods**. Clinical and pathological data of patients diagnosed with IgAN through biopsy at two centers were retrospectively collected. Logistic regression was employed to analyze the training cohort dataset and identify the independent predictors to construct a nomogram model based on the final variables. The predictive model was validated both internally and externally, with its performance assessed using the area under the curve (AUC), calibration curves, and decision curve analysis.

**Results**. Out of the patients in the modeling group, 129 individuals (41.6%) did not achieve remission following 3 months of treatment, indicating a high risk of CKD progression. A multivariate logistic regression analysis demonstrated that body mass index, urinary protein excretion, and tubular atrophy/interstitial fibrosis were identified as independent predictors for risk stratification. A nomogram model was formulated utilizing the final variables. The AUCs for the training set, internal validation set, and external validation set were 0.746 (95% confidence intervals (CI) [0.691–0.8]), 0.764 (95% CI [0.68–0.85]), and 0.749 (95% CI [0.65–0.85]), respectively. The validation of the subgroup analysis also demonstrated a satisfactory AUC.

**Conclusion**. This study developed and validated a practical nomogram that can individually predict short-term treatment outcomes ($\geq$90 days) and the risk of CKD progression in IgAN patients. It provides reliable guidance for timely and personalized intervention and treatment strategies.

## INTRODUCTION

IgA nephropathy (IgAN) is the most common pathological form of primary glomerulonephritis, which is predominantly found in children and young adults. It constitutes 45–60% of primary glomerular diseases and ranks among the primary causes of end-stage renal disease (ESRD) in China (*Hou et al., 2018*; *Stamellou et al., 2023*). Based on reliable statistics, approximately 30% to 40% of patients experience progression to end-stage renal disease 20 to 30 years after the initial onset of clinical symptoms (*Stamellou et al., 2023*). These patients require renal replacement therapy to prolong their lives, which imposes a great physical, mental, and economic burden on the patients and their families and has a serious impact on the social economy (*Lerma et al., 2023*). A systematic review related to outcome prediction in IgAN revealed that renal dysfunction, dialysis, and mortality have consistently been focal points of concern. Factors such as age of onset, obesity, hypertension, proteinuria, and the degree of histological pathology all contribute to the progression of IgAN (*Cattran, Floege & Coppo, 2023*). Unfortunately, few studies have focused on the short-term prognosis ($\geq 90$ days) of IgAN, although it may ultimately affect long-term outcomes.

Early disease progression risk prediction and stratification remain great challenges among treatment decisions for IgAN patients. Proteinuria stands out as the most crucial independent predictor of adverse renal outcomes, serving as a dependable surrogate endpoint and therapeutic target for forecasting long-term clinical consequences (*Thompson et al., 2019*). The 2021 Guidelines of the Kidney Disease: Improving Global Outcomes Committee (KDIGO) (*Beck et al., 2023*) recommend the following treatment objectives for IgAN: reducing urinary protein to below 1 g/d for at least 90 days with optimized supportive therapy. In addition, risk prediction and stratification (high risk, >0.75–1 g/d) for progression of chronic kidney disease (CKD) are recommended to guide the development of clinical treatment strategies. According to this authoritative clinical practice guideline, a critical time point has been identified: 90 days after diagnosis. This is because the treatment approach in subsequent stages will be determined based on the extent of urinary protein remission at this stage. For patients with urinary protein levels exceeding 1 g/day, maintenance or intensification of immunosuppressive therapy is recommended. However, it is important to acknowledge that this situation may expose patients to a higher risk of adverse drug reactions and disease progression. In clinical practice, a significant number of patients still fail to achieve remission of proteinuria (>1 g/d) at least 90 days after treatment (>1 g/d), requiring maintenance or intensification of therapy. This situation increases the risk of adverse drug reactions and disease advancement. Therefore, early detection and management are crucial in improving early outcomes and preventing disease progression.

In this study, the objective was to retrospectively assess clinicopathological data and identify risk factors correlated with the chronic progression of IgAN. Additionally, we aimed to develop a nomogram model to identify the high-risk group among IgAN patients at an early stage and facilitate the formulation of individualized treatment regimens, ultimately reducing the risk of poor renal prognosis.
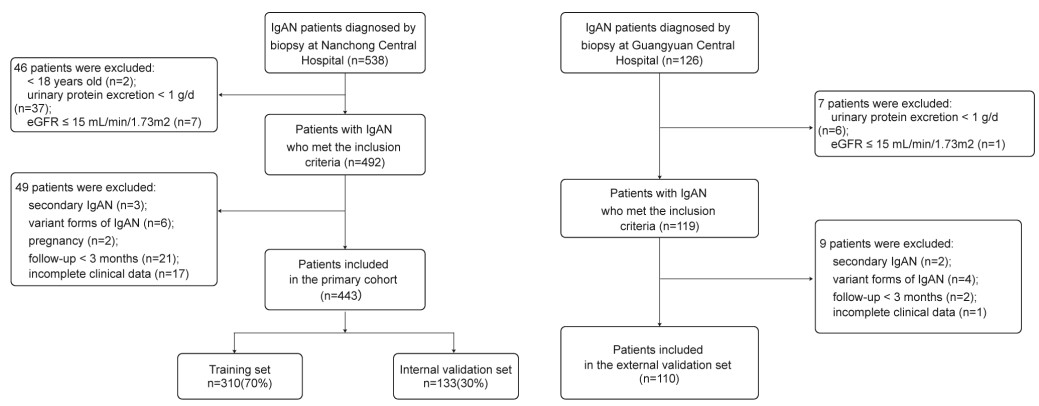

**Figure 1** **Recruitment process flowchart for research participants.** IgAN, IgA nephropathy.

# PARTICIPANTS AND METHODS

## Participants

The primary study cohort comprised data from 443 primary IgAN patients who underwent kidney biopsies at Nanchong Central Hospital from January 1, 2015, to December 31, 2023. This data was utilized for model development and internal validation. In addition, data from 110 primary IgAN patients who underwent kidney biopsies at Guangyuan Central Hospital within the same healthcare system between January 1, 2024, and May 31, 2024, were used for external validation. The study adhered to the principles outlined in the Declaration of Helsinki and received approval from the Ethics Committee of two centers. Oral informed consent was obtained from each patient; however, written consent was waived due to the retrospective nature of the study (No: 2023-100).

The inclusion and exclusion criteria were as follows. Inclusion criteria: (1) Age ≥ 18 years; (2) confirmed initial diagnosis of IgAN by renal biopsy; (3) urinary protein excretion (UPE) ≥ 1 g/d with estimated glomerular filtration rate (eGFR) >15 mL/min/1.73 m². The exclusion criteria were as follows: (1) Secondary IgAN, including lupus nephritis, allergic purpura nephritis, and hepatitis B virus-associated glomerulonephritis; (2) variant forms of IgAN, including IgA deposition with minimal change disease, acute kidney injury, and rapidly progressive glomerulonephritis; (3) pregnancy; (4) follow-up for <3 months and incomplete clinical data (Fig. 1).

## Outcome and variables

The outcome variable in this study was the stratification of risk for progression of CKD. Here, we defined the outcome variable according to the 2021 KDIGO guidelines (*Beck et al., 2023*) and as outlined in the study by *Canney et al. (2021)*. Low risk of CKD progression was defined as urinary protein <1 g/d and a decrease of at least 25% from baseline after renal biopsy and treatment for 90 days. High risk of CKD progression was defined as urinary protein ≥ 1 g/d after renal biopsy and treatment for 90 days, or a decrease in urinary protein less than 25% from baseline.

This was a cross-sectional survey that utilized baseline information from hospitalization records at the time of biopsy, and obtained through an electronic medical record system. The demographic and clinical data included sex, age, body mass index (BMI), systolic blood pressure (SBP), diastolic blood pressure (DBP), medical history, complications, and medication usage records. Laboratory data included neutrophil count, lymphocyte count, hemoglobin, platelet, serum albumin, serum creatinine, eGFR (calculated using the Chronic Kidney Disease Epidemiology Collaboration equation (*Levey, Inker & Coresh, 2014*)), serum uric acid, serum cystatin C, total cholesterol, triglycerides, serum high-sensitivity C-reactive protein (Hs-CRP), serum immunoglobulin and complement levels, and UPE. Renal biopsy data included Oxford Classification scores, which consist of mesangial hypercellularity (M): M0 ≤ 0.5, M1 > 0.5; endocapillary hypercellularity (E): E0 = absent, E1 = present; segmental glomerulosclerosis (S): S0 = absent, S1 = present; tubular atrophy/interstitial fibrosis (T): T0 < 25%, T1 = 25%–50%, T2 > 50%; and cellular/fibrocellular crescents (C): C0 = absent, C1 < 25%, C2 > 25% (*Trimarchi et al., 2017*). The histopathological findings of all renal biopsy specimens were independently evaluated by two pathology experts from the corresponding hospital. Decisions regarding supportive therapy and/or immunosuppressive treatment were made based on the judgment of the attending nephrology specialists.

### Development and validation of the nomogram

Before constructing the nomogram model, we randomly divided the patient data from the primary cohort into a training set and an internal validation set in a 7:3 ratio using R (dplyr 1.1.4). The nomogram was then developed using the training set data. Univariate logistic regression analysis and Forward Selection for Multivariate Logistic Regression were performed with SPSS 27.0 to identify independent predictors with statistical significance ($P < 0.05$). Subsequently, BMI, UPE, and T were used to construct the nomogram.

Internal and external validation were conducted on the nomogram. Model predictive accuracy was evaluated using 1,000 bootstrap resamples in R (rms 6.8.1). Receiver operating characteristic (ROC) curves were generated with R (pROC 1.18.5) to assess the model's discriminative performance, while decision curve analysis (DCA) curves were created using R (rmda 1.6) to evaluate the model's clinical utility. Additionally, we conducted a subgroup analysis based on different treatment regimens.

### Statistical analysis

Statistical analysis was executed utilizing SPSS 27.0, R 4.4.1. Continuous variables were assessed for normality using the Shapiro–Wilk test. Quantitative characteristics that were normally distributed were described as means and standard deviations, while those not normally distributed were reported as the median and interquartile range (IQR), and differences between groups were assessed through the Wilcoxon rank sum test. Qualitative characteristics were described as frequencies (percentages), and differences between groups were assessed through Pearson's Chi-squared test or Fisher's exact test. Logistic regression was conducted on the training set data to screen for key influencing factors, and the nomogram was generated using R software. The predictive capability of the model was

assessed through the area under curve (AUC), calibration curve, and DCA. $P < 0.05$ was defined as a statistically significant difference.

## RESULTS

### Baseline characteristics

With a rigorous inclusion and exclusion screening process, the study ultimately enrolled 553 primary IgAN patients, who were divided into 310 patients for the training set, 133 for the internal validation set, and 110 for the external validation set. Among them, 213 patients (38.5%) did not achieve remission of proteinuria at least 90 days of treatment, indicating a high risk for CKD progression. The study population predominantly consisted of young and middle-aged women, with an average age of 36 (28–48) years, and 56.2% of the patients were female. Specific baseline data for the study population are detailed in Table 1.

In the training set, patients in the high-risk group exhibited significantly higher levels of BMI, DBP, neutrophil count, serum creatinine, serum cystatin C, cholesterol, triglycerides, hs-CRP, serum C3, UPE and hematuria compared to those in the low-risk group. Furthermore, the high-risk group demonstrated a higher prevalence of diabetes, CKD3-4 stage and nephrotic syndrome. They also displayed more severe renal tubular atrophy/interstitial fibrosis and greater formation of cellular/fibrocellular crescents. Meanwhile, the high-risk group had lower levels of albumin, eGFR and serum IgG (Table 2).

### Identification of risk factors

Univariate logistic regression analysis was applied to analyze the baseline variables of the training set. The results showed that 17 variables, including BMI, DBP, neutrophil count, albumin, serum creatinine, eGFR, serum cystatin C, triglycerides, serum IgG, serum C3, UPE, hematuria, CKD stage, nephrotic syndrome, diabetes, T, and C, were statistically significant factors ($P < 0.05$). Multivariate logistic regression analysis was then executed on those characteristics (Table 3).

The multivariate logistic regression model identified three risk factors independently correlated with CKD progression risk: BMI (OR = 1.109, 95% CI [1.042–1.182], $P = 0.001$), UPE (OR = 1.502, 95% CI [1.268–1.779], $P < 0.001$), and T1/2 (OR = 2.134, 95% CI [1.226–3.714], $P = 0.007$). The remaining factors did not show significant statistical significance ($P > 0.05$, Table 3).

### Establishment and validation of the nomogram

A nomogram was created using the results of the multivariate logistic regression analysis, which identified three risk factors. This nomogram visualizes the practical use of the model using a random sample (Fig. 2). The nomogram model assigns scores to each independent variable based on their contribution to the outcome event (as indicated by the magnitude of regression coefficients). These scores are then summed to obtain a total score, which is converted into a probability score. Higher scores indicate a higher risk of CKD progression. An example of the nomogram applied to a random sample is illustrated in Fig. 2.
**Table 1  Baseline characteristics of the study cohort.**

| Variables | Overall (*N* = 553) | Training set (*N* = 310) | Internal valida- tion set (*N* = 133) | External valida- tion set (*N* = 110) | *P* value |
|---|---|---|---|---|---|
| Non-remission, n (%) | 213 (38.5) | 129 (41.6) | 46 (34.6) | 38 (34.5) | 0.240 |
| Sex (male, %) | 242 (43.8) | 141 (45.5) | 64 (48.1) | 37 (33.6) | 0.050 |
| Age (years) | 36.000 (28.000, 48.000) | 35.000 (27.250, 47.000) | 40.000 (28.000, 50.000) | 38.000 (29.000, 46.000) | 0.245 |
| BMI (kg/m$^2$) | 23.880 (21.600, 26.000) | 23.950 (21.405, 26.303) | 24.030 (22.170, 26.080) | 23.375 (20.848, 25.400) | 0.073 |
| SBP (mmHg) | 131.000 (117.000, 144.000) | 130.000 (115.000, 146.000) | 132.500 (120.250, 144.750) | 131.500 (118.250, 140.000) | 0.448 |
| DBP (mmHg) | 85.000 (76.000, 95.000) | 85.000 (76.000, 95.000) | 86.000 (77.000, 95.750) | 84.000 (75.250, 92.000) | 0.560 |
| Hypertension, n (%) | | | | | <0.001 |
| NO | 264 (48.7) | 138 (45.7) | 53 (40.8) | 73 (66.4) | |
| Yes | 278 (51.3) | 164 (54.3) | 77 (59.2) | 37 (33.6) | |
| Diabetes, n (%) | | | | | 0.020 |
| NO | 494 (89.3) | 269 (86.8) | 119 (89.5) | 106 (96.4) | |
| Yes | 59 (10.7) | 41 (13.2) | 14 (10.5) | 4 (3.64) | |
| CKD stage, n (%) | | | | | 0.004 |
| Stage 1-2 | 439 (79.5) | 238 (77.0) | 101 (75.9) | 100 (90.9) | |
| Stage 3-4 | 113 (20.5) | 71 (23.0) | 32 (24.1) | 10 (9.09) | |
| Nephrotic syndrome, n (%) | | | | | 0.306 |
| NO | 513 (92.8) | 292 (94.2) | 120 (90.2) | 101 (91.8) | |
| Yes | 40 (7.23) | 18 (5.81) | 13 (9.77) | 9 (8.18) | |
| Neutrophil ($\times 10^9$/L) | 4.390 (3.440, 5.480) | 4.440 (3.450, 5.810) | 4.360 (3.300, 5.130) | 4.370 (3.573, 5.445) | 0.501 |
| Lymphocyte ($\times 10^9$/L) | 1.590 (1.230, 2.000) | 1.605 (1.240, 2.030) | 1.670 (1.240, 2.010) | 1.535 (1.193, 1.915) | 0.413 |
| Hemoglobin (g/L) | 127.602 (21.606) | 128.513 (21.595) | 125.444 (21.544) | 127.645 (21.725) | 0.349 |
| Platelet ($\times 10^9$/L) | 202.000 (157.000, 247.000) | 209.500 (163.250, 255.750) | 200.000 (157.000, 242.000) | 185.500 (143.250, 223.500) | 0.006 |
| Albumin (g/L) | 38.800 (35.400, 42.400) | 38.700 (35.540, 42.575) | 38.900 (35.300, 42.700) | 39.300 (35.100, 41.675) | 0.879 |
| Serum creatinine (umol/L) | 83.600 (63.950, 105.175) | 84.000 (64.000, 108.600) | 89.000 (65.000, 109.300) | 76.500 (62.000, 94.000) | 0.010 |
| eGFR (ml/min/1.73 m2) | 88.440 (65.163, 111.565) | 88.540 (62.870, 112.180) | 82.320 (61.870, 105.470) | 97.025 (73.798, 115.843) | 0.011 |

| Variables | Overall (N = 553) | Training set (N = 310) | Internal valida-tion set (N = 133) | External valida-tion set (N = 110) | P value |
|---|---|---|---|---|---|
| Serum uric acid (umol/L) | 362.100 (297.925, 434.000) | 362.500 (291.600, 440.000) | 378.900 (319.600, 434.800) | 348.000 (293.250, 414.250) | 0.114 |
| Serum cystatin C (mg/L) | 1.100 (0.890, 1.470) | 1.070 (0.883, 1.643) | 1.140 (0.890, 1.590) | 1.070 (0.940, 1.280) | 0.416 |
| Total cholesterol (mmol/L) | 4.900 (4.190, 5.690) | 4.890 (4.140, 5.810) | 4.930 (4.213, 5.780) | 4.870 (4.315, 5.235) | 0.714 |
| Triglycerides (mmol/L) | 1.640 (1.090, 2.360) | 1.560 (1.018, 2.293) | 1.550 (1.075, 2.455) | 1.750 (1.295, 2.278) | 0.508 |
| Hs-CRP (mg/L) | 2.000 (0.650, 4.500) | 2.125 (0.635, 4.860) | 1.300 (0.600, 4.370) | 2.165 (0.793, 3.980) | 0.321 |
| Serum IgG (g/L) | 10.120 (8.060, 12.240) | 9.985 (7.955, 12.100) | 10.300 (7.840, 13.040) | 10.460 (8.445, 12.178) | 0.381 |
| Serum IgM (g/L) | 1.240 (0.890, 1.760) | 1.180 (0.854, 1.717) | 1.220 (0.870, 1.850) | 1.360 (1.000, 1.740) | 0.320 |
| Serum IgA (g/L) | 2.920 (2.340, 3.630) | 2.995 (2.350, 3.728) | 2.720 (2.200, 3.670) | 2.910 (2.453, 3.403) | 0.311 |
| Serum C3 (g/L) | 0.940 (0.810, 1.080) | 0.903 (0.780, 1.050) | 0.884 (0.798, 1.042) | 1.045 (0.940, 1.150) | <0.001 |
| Serum C4 (g/L) | 0.260 (0.207, 0.326) | 0.261 (0.204, 0.321) | 0.254 (0.201, 0.327) | 0.260 (0.224, 0.330) | 0.587 |
| UPE (g/d) | 1.910 (1.430, 3.060) | 1.950 (1.473, 3.143) | 2.140 (1.660, 3.220) | 1.445 (1.195, 2.190) | <0.001 |
| Hematuria (RBCs/ul) | 68.500 (12.000, 198.000) | 88.000 (15.750, 252.250) | 90.500 (17.750, 241.000) | 16.000 (5.000, 69.000) | <0.001 |
| **Oxford classification, n (%)** | | | | | |
| M1 | 536 (96.9) | 297 (95.8) | 129 (97.0) | 110 (100.0) | 0.064 |
| E1 | 215 (38.9) | 126 (40.6) | 49 (36.8) | 40 (36.4) | 0.628 |
| S1 | 310 (56.1) | 202 (65.2) | 75 (56.4) | 33 (30.0) | <0.001 |
| T1/2 | 181 (32.7) | 106 (34.2) | 49 (36.8) | 26 (23.6) | 0.065 |
| C1/2 | 197 (35.6) | 124 (40.0) | 62 (46.6) | 11 (10.0) | <0.001 |
| **Treatment, n (%)** | | | | | 0.753 |
| RAASi alone | 200 (36.2) | 112 (36.1) | 47 (35.3) | 41 (37.3) | |
| RAASi + Glucocorticoid | 175 (31.6) | 100 (32.3) | 37 (27.8) | 38 (34.5) | |
| RAASi + Glucocorticoid + Immunosuppressant | 60 (10.8) | 30 (9.68) | 18 (13.5) | 12 (10.9) | |
| Others | 118 (21.3) | 68 (21.9) | 31 (23.3) | 19 (17.3) | |

**Notes.**

Kruskal–Wallis rank sum test or Pearson's Chi-squared test or Fisher's exact test was used for comparison between groups.

BMI, body mass index; SBP, systolic blood pressure; DBP, diastolic blood pressure; CKD, chronic kidney disease; eGFR, estimated glomerular filtration rate; hs-CRP, high-sensitivity C-reactive protein; UPE, urinary protein excretion; RBC, red blood cells; M, mesangial hypercellularity; E, endocapillary hypercellularity; S, segmental glomeruloscle-rosis; T, interstitial fibrosis/tubular atrophy; C, crescent formation; RAASi, renin-angiotensin-aldosterone system inhibitors, including angiotensin converting enzyme inhibitor and angiotensin receptor blocker; Immunosuppressant included Cytoxan, mycophenolate mofetil, Cyclosporine, Tacrolimus.
**Table 2  Baseline characteristics of the training set.**

| Variables | Overall ($N = 310$) | Low risk ($N = 181$) | High risk ($N = 129$) | *P* value |
|---|---|---|---|---|
| Sex (male, %) | 141 (45.5) | 80 (44.2) | 61 (47.3) | 0.590 |
| Age (years) | 35.000 (27.250, 47.000) | 34.000 (27.000, 46.000) | 36.000 (28.000, 49.000) | 0.177 |
| BMI (kg/m$^2$ ) | 23.950 (21.405, 26.303) | 22.890 (20.700, 25.530) | 24.910 (22.950, 27.480) | <0.001 |
| SBP (mmHg) | 130.000 (115.000, 146.000) | 129.500 (114.750, 144.250) | 131.000 (118.000, 146.000) | 0.331 |
| DBP (mmHg) | 85.000 (76.000, 95.000) | 83.000 (75.000, 93.250) | 88.000 (79.000, 97.000) | 0.013 |
| Hypertension, n (%) | | | | 0.112 |
| NO | 138 (45.7) | 89 (49.4) | 49 (40.2) | |
| Yes | 164 (54.3) | 91 (50.6) | 73 (59.8) | |
| Diabetes, n (%) | | | | 0.018 |
| NO | 269 (86.8) | 164 (90.6) | 105 (81.4) | |
| Yes | 41 (13.2) | 17 (9.39) | 24 (18.6) | |
| CKD stage, n (%) | | | | <0.001 |
| Stage 1-2 | 238 (77.0) | 152 (84.4) | 86 (66.7) | |
| Stage 3-4 | 71 (23.0) | 28 (15.6) | 43 (33.3) | |
| Nephrotic syndrome, n (%) | | | | <0.001 |
| NO | 292 (94.2) | 179 (98.9) | 113 (87.6) | |
| Yes | 18 (5.81) | 2 (1.10) | 16 (12.4) | |
| Neutrophil ($\times 10^9$/L) | 4.440 (3.450, 5.810) | 4.250 (3.360, 5.470) | 4.690 (3.590, 6.470) | 0.026 |
| Lymphocyte ($\times 10^9$/L) | 1.605 (1.240, 2.030) | 1.560 (1.220, 2.030) | 1.610 (1.310, 2.030) | 0.291 |
| Hemoglobin (g/L) | 128.513 (21.595) | 128.895 (19.367) | 127.977 (24.450) | 0.756 |
| Platelet ($\times 10^9$/L) | 209.500 (163.250, 255.750) | 214.000 (168.000, 252.000) | 205.000 (155.000, 263.000) | 0.596 |
| Albumin (g/L) | 38.700 (35.540, 42.575) | 39.900 (37.000, 43.300) | 37.500 (33.500, 41.200) | <0.001 |
| Serum creatinine (umol/L) | 84.000 (64.000, 108.600) | 79.450 (63.150, 103.550) | 88.900 (67.000, 133.200) | 0.028 |
| eGFR (ml/min/1.73 m2) | 88.540 (62.870, 112.180) | 90.060 (72.403, 112.943) | 85.560 (47.210, 110.490) | 0.028 |
| Serum Uric acid (umol/L) | 362.500 (291.600, 440.000) | 360.400 (288.100, 435.650) | 368.600 (292.200, 443.700) | 0.564 |
| Serum cystatin C (mg/L) | 1.070 (0.883, 1.643) | 1.030 (0.820, 1.360) | 1.140 (0.900, 1.920) | 0.027 |
| Total cholesterol (mmol/L) | 4.890 (4.140, 5.810) | 4.670 (4.030, 5.560) | 5.315 (4.513, 6.028) | <0.001 |
| Triglycerides (mmol/L) | 1.560 (1.018, 2.293) | 1.440 (0.990, 1.978) | 1.805 (1.130, 2.873) | 0.003 |
| Hs-CRP (mg/L) | 2.125 (0.635, 4.860) | 1.210 (0.550, 4.450) | 3.230 (0.870, 5.380) | 0.022 |
| Serum IgG (g/L) | 9.985 (7.955, 12.100) | 10.490 (8.460, 12.300) | 9.300 (7.490, 11.500) | 0.006 |
| Serum IgM (g/L) | 1.180 (0.854, 1.717) | 1.270 (0.950, 1.700) | 1.110 (0.780, 1.720) | 0.147 |
| Serum IgA (g/L) | 2.995 (2.350, 3.728) | 3.090 (2.430, 3.750) | 2.760 (2.260, 3.660) | 0.056 |
| Serum C3 (g/L) | 0.903 (0.780, 1.050) | 0.888 (0.780, 1.020) | 0.950 (0.792, 1.080) | 0.034 |
| Serum C4 (g/L) | 0.261 (0.204, 0.321) | 0.250 (0.200, 0.310) | 0.270 (0.210, 0.346) | 0.126 |

**Table 2** (*continued*)

| Variables | Overall (N = 310) | Low risk (N = 181) | High risk (N = 129) | P value |
|---|---|---|---|---|
| UPE (g/d) | 1.950 (1.473, 3.143) | 1.700 (1.380, 2.370) | 2.780 (1.760, 5.010) | <0.001 |
| Hematuria (RBCs/ul) | 88.000 (15.750, 252.250) | 73.500 (11.000, 180.000) | 124.000 (52.250, 385.000) | 0.002 |
| **Oxford classification, n (%)** | | | | |
| M1 | 297 (95.8) | 173 (95.6) | 124 (96.1) | 0.814 |
| E1 | 126 (40.6) | 67 (37.0) | 59 (45.7) | 0.123 |
| S1 | 202 (65.2) | 116 (64.1) | 86 (66.7) | 0.639 |
| T1/2 | 106 (34.2) | 45 (24.9) | 61 (47.3) | <0.001 |
| C1/2 | 124 (40.0) | 62 (34.3) | 62 (48.1) | 0.014 |
| **Treatment, n (%)** | | | | 0.907 |
| RAASi alone | 112 (36.1) | 66 (36.5) | 46 (35.7) | |
| RAASi + Glucocorticoid | 100 (32.3) | 56 (30.9) | 44 (34.1) | |
| RAASi + Glucocorticoid + Immunosuppressant | 30 (9.68) | 19 (10.5) | 11 (8.53) | |
| Others | 68 (21.9) | 40 (22.1) | 28 (21.7) | |

**Notes.**

Wilcoxon rank sum test or Pearson's Chi-squared test was used for comparison between groups.

BMI, body mass index; SBP, systolic blood pressure; DBP, diastolic blood pressure; CKD, chronic kidney disease; eGFR, estimated glomerular filtration rate; hs-CRP, high-sensitivity C-reactive protein; UPE, urinary protein excretion; RBC, red blood cells; M, mesangial hypercellularity; E, endocapillary hypercellularity; S, segmental glomerulosclerosis; T, interstitial fibrosis/tubular atrophy; C, crescent formation; RAASi, renin-angiotensin-aldosterone system inhibitors, including angiotensin converting enzyme inhibitor and angiotensin receptor blocker; Immunosuppressant included Cytoxan, mycophenolate mofetil, Cyclosporine, Tacrolimus.

We employed R (pROC 1.18.5) and R (rms 6.8.1) to generate ROC curves and calibration curves for evaluating the predictive performance of the model on the training set and validation set. The results demonstrate that the nomogram model exhibits good discriminative ability, with AUCs of 0.746 (95% CI [0.691–0.8]), 0.764 (95% CI [0.68–0.85]), and 0.749 (95% CI [0.65–0.85]) for the training set, internal validation set, and external validation set, respectively; subgroup analysis validation based on treatment measures also showed a satisfied AUC, as depicted in Fig. 3.

The calibration curves fitted well with the ideal curve in the training set and the internal and external validation sets, indicating a high level of consistency between the predicted probabilities and actual event rates of the model, as shown in Fig. 4.

The model demonstrated good predictive accuracy in discrimination and calibration; however, its clinical utility remained uncertain. To validate the clinical applicability of the model, we plotted DCA curves using R (rmda 1.6). The DCA curves revealed that the net clinical benefit of the model decreased as the threshold probability increased. Specifically, when the threshold probability values ranged from 1.0% to 88.0% for the training set, and 1.0% to 68.0% for the internal validation set, and 1.0% to 95% for the external validation set, the model provided a net benefit that surpassed both "treat all" or "treat none" strategies, as depicted in Fig. 5.

## DISCUSSION

Our study discovered that even with proactive treatment at least 90 days after diagnosis, 38.5% of IgAN patients did not experience remission in proteinuria, requiring the continuation or intensification of therapy. The finding indicates a high prevalence of

**Table 3 Logistic regression assessing risk factors for CKD progression risk.**

| Variables | Univariable analysis | | | Multivariable analysis | | |
|---|---|---|---|---|---|---|
| | *OR* | 95%*CI* | *P* value | *OR* | 95%*CI* | *P* value |
| Sex (male) | 1.133 | 0.720–1.782 | 0.591 | | | |
| Age (years) | 1.013 | 0.995–1.03 | 0.151 | | | |
| BMI (kg/m$^2$) | 1.103 | 1.04–1.169 | 0.001 | 1.109 | 1.042–1.182 | 0.001 |
| SBP (mmHg) | 1.004 | 0.993–1.014 | 0.472 | | | |
| DBP (mmHg) | 1.021 | 1.003–1.038 | 0.019 | | | |
| Hypertension | 1.457 | 0.915–2.320 | 0.113 | | | |
| Diabetes | 2.205 | 1.131–4.300 | 0.02 | | | |
| CKD stage | | | | | | |
| Stage 1–2 | Ref | Ref | Ref | | | |
| Stage 3–4 | 2.603 | 1.517–4.469 | 0.001 | | | |
| Nephrotic syndrome | 12.673 | 2.860–56.155 | 0.001 | | | |
| Neutrophil ($\times10^9$/L) | 1.161 | 1.05–1.283 | 0.004 | | | |
| Lymphocyte ($\times10^9$/L) | 1.18 | 0.853–1.634 | 0.318 | | | |
| Hemoglobin (g/L) | 0.998 | 0.988–1.009 | 0.712 | | | |
| Platelet ($\times10^9$/L) | 1 | 0.997–1.003 | 0.925 | | | |
| Albumin (g/L) | 0.921 | 0.883–0.96 | <0.001 | | | |
| Serum creatinine (umol/L) | 1.006 | 1.002–1.01 | 0.006 | | | |
| eGFR (ml/min/1.73 m2) | 0.991 | 0.984–0.998 | 0.014 | | | |
| Serum Uric acid (umol/L) | 1.001 | 0.999–1.003 | 0.556 | | | |
| Serum cystatin C (mg/L) | 1.616 | 1.15–2.271 | 0.006 | | | |
| Total cholesterol (mmol/L) | 1.326 | 1.113–1.579 | 0.002 | | | |
| Triglycerides (mmol/L) | 1.114 | 0.984–1.262 | 0.088 | | | |
| Hs-CRP (mg/L) | 0.994 | 0.974–1.015 | 0.567 | | | |
| Serum IgG (g/L) | 0.901 | 0.83–0.978 | 0.013 | | | |
| Serum IgM (g/L) | 0.92 | 0.672–1.26 | 0.603 | | | |
| Serum IgA (g/L) | 0.817 | 0.645–1.034 | 0.093 | | | |
| Serum C3 (g/L) | 5.155 | 1.557–17.067 | 0.007 | | | |
| Serum C4 (g/L) | 11.279 | 0.931–136.657 | 0.057 | | | |
| UPE (g/d) | 1.561 | 1.332–1.829 | <0.001 | 1.502 | 1.268–1.779 | <0.001 |
| Hematuria (RBCs/ul) | 1.001 | 1.0–1.001 | 0.032 | | | |
| **Oxford classification** | | | | | | |
| M1 | 1.147 | 0.366–3.589 | 0.814 | | | |
| E1 | 1.434 | 0.906–2.270 | 0.124 | | | |
| S1 | 1.121 | 0.697–1.803 | 0.639 | | | |
| T1/2 | 2.711 | 1.673–4.394 | <0.001 | 2.134 | 1.226–3.714 | 0.007 |
| C1/2 | 1.776 | 1.119–2.819 | 0.015 | | | |
| **Treatment** | | | | | | |
| RAASi alone | 0.996 | 0.540–1.837 | 0.989 | | | |
| RAASi + Glucocorticoid | 1.122 | 0.602–2.095 | 0.717 | | | |
| RAASi + Glucocorticoid + Immunosuppressant | 0.827 | 0.341–2.006 | 0.674 | | | |

**Table 3** (*continued*)

| Variables | Univariable analysis | | | Multivariable analysis | | |
|---|---|---|---|---|---|---|
| | *OR* | 95%*CI* | *P* value | *OR* | 95%*CI* | *P* value |
| Others | Ref | Ref | Ref | | | |

**Notes.**

*OR*, odds ratio; *CI*, confidence interval; BMI, body mass index; SBP, systolic blood pressure; DBP, diastolic blood pressure; CKD, chronic kidney disease; eGFR, estimated glomerular filtration rate; hs-CRP, high-sensitivity C-reactive protein; UPE, urinary protein excretion; RBC, red blood cells; M, mesangial hypercellularity; E, endocapillary hypercellularity; S, segmental glomerulosclerosis; T, interstitial fibrosis/tubular atrophy; C, crescent formation. RAASi, renin-angiotensin-aldosterone system inhibitors, including angiotensin converting enzyme inhibitor and angiotensin receptor blocker; Immunosuppressant included Cytoxan, mycophenolate mofetil, Cyclosporine, Tacrolimus.

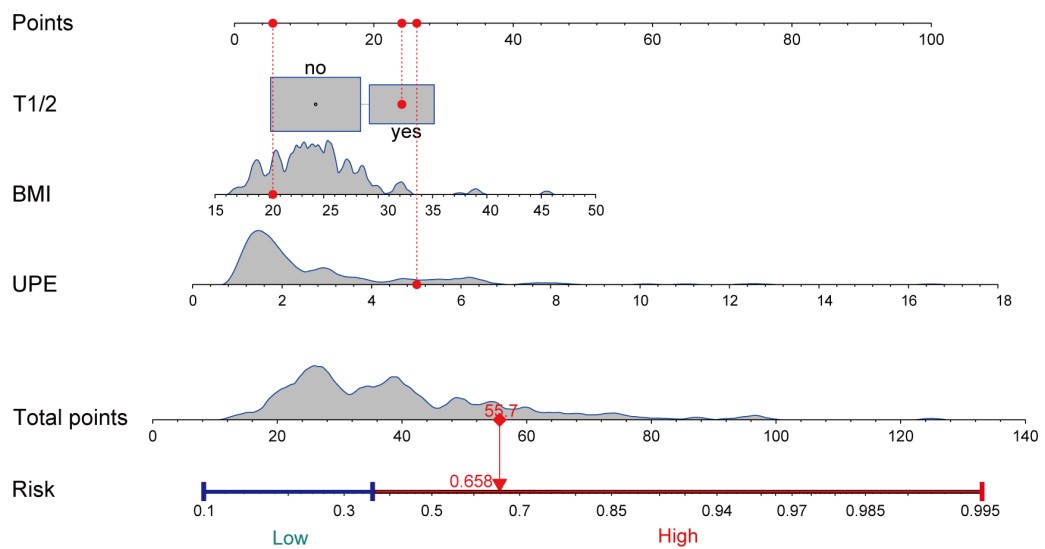

**Figure 2 Nomogram to estimate the risk of CKD progression in IgAN patients.** We visualized the application of the nomogram model based on clinical cases (highlighted in red). Each predictor in the case receives a corresponding "Points", and summed to obtain a "Total points", and plotting a vertical line downward yields a corresponding risk value to determine the risk of CKD progression. T, tubular atrophy/interstitial fibrosis; BMI, body mass index; UPE, urinary protein excretion.

patients at risk for CKD progression in clinical practice, highlighting the need for increased attention to this issue. Further univariate and multivariate logistic regression analysis indicates that the lack of reduction in proteinuria in these patients may be closely associated with BMI, UPE, and T1/2. Significantly, we developed a nomogram model based on these three variables and performed both internal and external validation. The results show that the nomogram model demonstrates good discriminative ability, predictive accuracy, and clinical utility in both the training set and the validation set.

BMI is the primary indicator of weight assessment in clinical practice worldwide. *Yun et al. (2018)* found that obesity, whether accompanied by metabolic disorders or not, can lead to progression and is an independent risk factor for the deterioration of CKD. *Bonnet et al. (2001)* first proposed in 2001 that obesity was a novel independent risk factor for the clinical and pathological progression of primary IgAN, which was subsequently corroborated by multiple studies (*Berthoux, Mariat & Maillard, 2013*; *Shimamoto et al., 2015*; *Wu, Wang & Li, 2018*; *Kataoka et al., 2012*). Recently, a meta-analysis has shown that higher BMI in

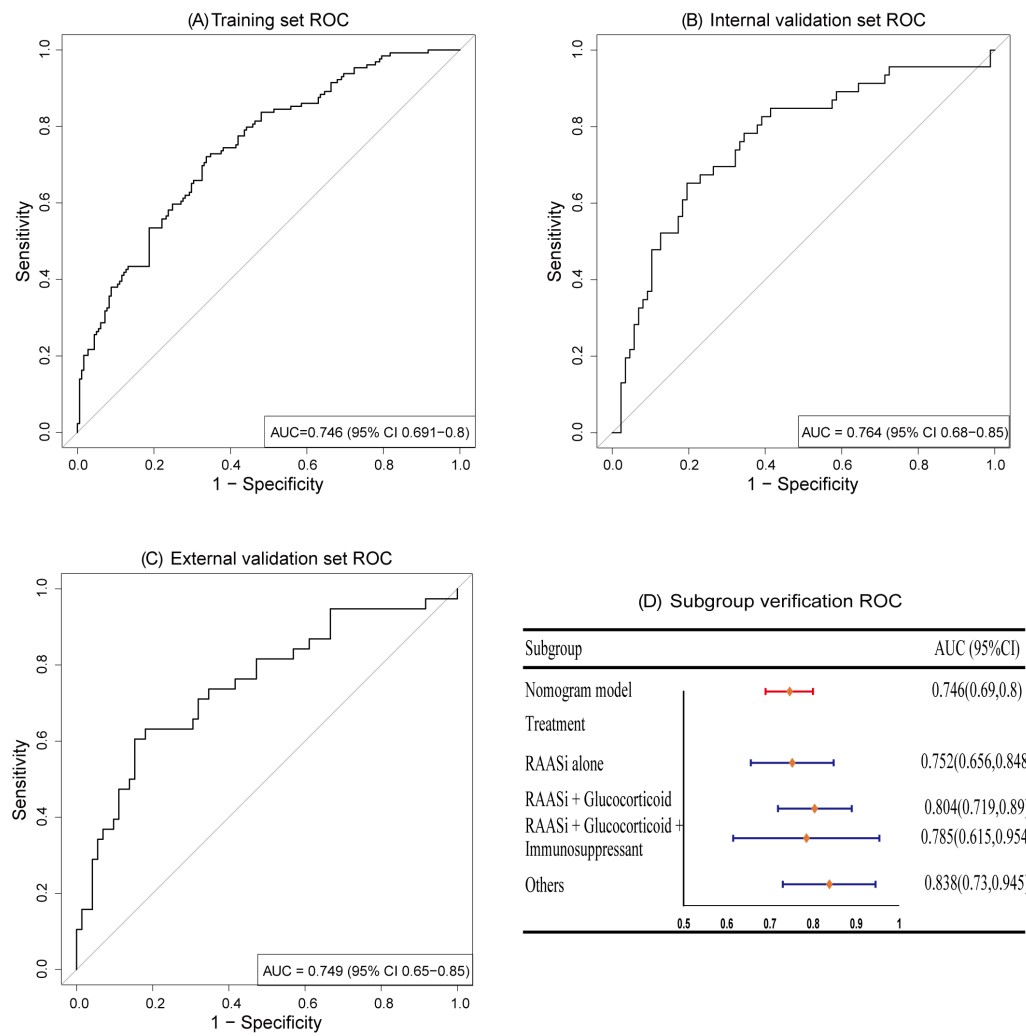

**Figure 3 ROC curves.** The AUCs of training set (A), internal validation set (B), and external validation set (C) showed that the model has a good discrimination ability. Subgroup validation of AUCs based on treatment regimens (D).

IgAN patients might be associated with lower kidney function (*Kanbay et al., 2022*). IgAN patients complicated with obesity had more severe renal dysfunction at the time of renal biopsy than those with optimal body weight (*Wang et al., 2023*). In our research, we found that a high BMI is a risk factor for CKD progression in patients with IgAN. Notably, our study had a limited observation period of three months and minimal fluctuations in BMI, which enhances the reliability of our findings. Excessive weight can potentially trigger or exacerbate renal pathological damage by affecting intrarenal hemodynamics (increased renal blood flow and hyperfiltration), accelerating the progression of kidney disease in patients. Meanwhile, it was reported that high BMI indirectly accelerated the progression of IgAN by inducing metabolic syndrome (*Kataoka et al., 2012*). Furthermore, consistent with the findings of previous studies, our research also highlights baseline proteinuria as a

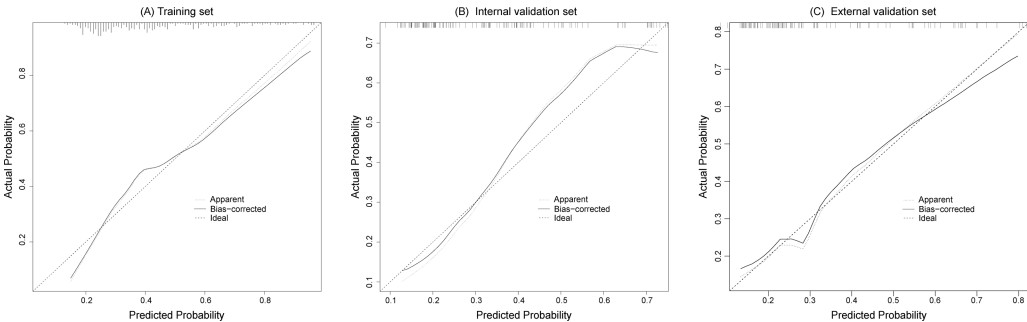

**Figure 4** The calibration curves of the training set (A), internal validation set (B) and external validation set (C) both fit well with the ideal curve, demonstrating good consistency between predicted and actual risk probability.

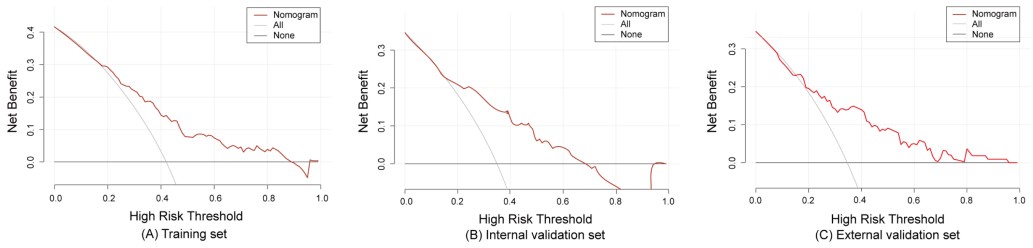

**Figure 5** The decision curve analysis of the training set (A), internal validation set (B) and external validation set (C).

significant risk factor influencing the progression of CKD in patients with IgAN (*Gadola et al., 2023*; *Chen et al., 2018*).

Several studies have explored the relationship between Oxford classification indicators and the prognosis of IgAN, with T1/2 holding the most significant prognostic value. The results of the VALIGA study, which followed 1,147 IgAN patients from 13 European countries to assess the predictive value of the Oxford classification for renal outcomes, revealed that T1/2 lesions serve as independent predictors of a 50% decrease in eGFR or ESRD (*Coppo et al., 2014*). This conclusion was similarly supported by the study after 4 years of updated follow-up (*Coppo et al., 2020*). Similarly, *Chen, Wu & Tsai (2023)* found that T1/2 was the most predictive variable for renal prognosis (AUC = 0.73), consistently correlating with poorer renal outcomes across all subgroups and baseline states. Our research also identified that T1/2 is a predictor of short-term prognosis and disease progression in patients with IgAN (*Tang et al., 2023*). Strikingly, a recent systematic evaluation indicated that T1/2 was the Oxford element most frequently associated with IgAN outcomes (*Howie & Lalayiannis, 2023*). Given that T1/2 scoring is more objective and exhibits better reproducibility, it largely reflects a chronic pathological state that cannot be reversed through treatment at the time of biopsy, as well as a higher impact of tubular injury on renal function compared to glomerular injury. This could explain the significant association between T-scoring and disease progression (*Howie & Lalayiannis, 2023*).

IgAN is the most frequent primary glomerular nephritis and one of the leading reasons of ESRD (*Hou et al., 2018*). While numerous risk models have been developed to predict the progression of IgAN, these models often focus on ESRD or eGFR decline as the observed endpoints, which tends to overlook the importance of short-term efficacy (*Barbour et al., 2019*; *Schena et al., 2021*). To the best of our knowledge, this is the first nomogram model to predict treatment response and risk of CKD progression after 90 days of diagnosis and treatment in patients with IgAN. Furthermore, it has been externally validated. However, there are some limitations to this study. Firstly, it was retrospective study, where all baseline data were derived from renal biopsy patients. This overlooks patients with milder conditions who did not undergo renal biopsy, which may introduce selection bias. Secondly, although our model has been externally validated, it is worth noting that the model was developed using a relatively small sample size and both the development cohort and validation cohort were from a single province in China, which may affect the generalizability and applicability of our findings to a broader population. Furthermore, this limited our full consideration of certain variables (*e.g.*, basement membrane thickness, degree of podocyte fusion, and proportion of glomerulosclerosis in the analysis of pathologic biopsies) due to the exposure to some missing data. These unconsidered variables may affect the accuracy of the outcome indicators. Therefore, future research should incorporate additional variables and larger sample sizes. Additionally, conducting prospective and multicenter study is necessary to further validate the factors identified in our analysis that influence disease progression in primary IgAN patients.

## CONCLUSION

In summary, this study found that IgAN patients with poor short-term efficacy ($\geq$90 days and UPE $\geq$ 1 g/d) and a high risk of CKD progression accounted for a high proportion of the overall population. Through the nomogram predictive model, high-risk individuals for CKD progression among IgAN patients can be better identified, enabling early intervention to halt disease progression. These findings carry important implications for guiding personalized clinical interventions.

## ACKNOWLEDGEMENTS

We express our gratitude to the team at Extreme Smart Analysis for their assistance. Additionally, our sincere appreciation goes to the Science and Technology Innovation Center at North Sichuan Medical College for its robust work platform.

### Funding

This research was supported by the Sichuan Provincial Department of Science and Technology Research Special Fund (2021YFS0259) and Nanchong Science and Technology Plan Project (22JCYJPT0005). The funders had no role in study design, data collection and analysis, decision to publish, or preparation of the manuscript.

## Grant Disclosures

The following grant information was disclosed by the authors:

The Sichuan Provincial Department of Science and Technology Research Special Fund: 2021YFS0259.

Nanchong Science and Technology Plan Project: 22JCYJPT0005.

## Competing Interests

The authors declare there are no competing interests.

## Author Contributions

- Ying Zhang conceived and designed the experiments, performed the experiments, analyzed the data, prepared figures and/or tables, authored or reviewed drafts of the article, and approved the final draft.
- Zhixin Wang performed the experiments, prepared figures and/or tables, and approved the final draft.
- Wenwu Tang analyzed the data, prepared figures and/or tables, authored or reviewed drafts of the article, and approved the final draft.
- Xinzhu Yuan analyzed the data, prepared figures and/or tables, and approved the final draft.
- Xisheng Xie conceived and designed the experiments, authored or reviewed drafts of the article, and approved the final draft.

## Human Ethics

The following information was supplied relating to ethical approvals (i.e., approving body and any reference numbers):

The study adhered to the principles outlined in the Declaration of Helsinki and received approval from the Ethics Committee of Nanchong Central Hospital and Guangyuan Central Hospital. Oral informed consent was obtained from each patient; however, written consent was waived due to the retrospective nature of the study (No: 2023-100).

## Data Availability

The raw data and R code are available at GitHub and Zenodo:

- https://github.com/zhangy-hub/Raw-data-and-code-files/tree/v1.0.0

- zhangy-hub. (2024). zhangy-hub/Raw-data-and-code-files: Raw data and code files (v1.0.0). Zenodo. https://doi.org/10.5281/zenodo.13624981.

## Supplemental Information

Supplemental information for this article can be found online at http://dx.doi.org/10.7717/peerj.18416#supplemental-information.

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
