# Peer review of "Development and internal and external validation of a nomogram model for predicting the risk of chronic kidney disease progression in IgA nephropathy patients"

_PeerJ, doi:10.7717/peerj.18416_

## Round 0.1 · original submission · Major Revisions

· Academic Editor

Major Revisions

Thank you for your submission. Before we can consider this manuscript for publication the following issues must be addressed: 1) data and code availability to reproduce the analyses must be provided with a DOI (https://peerj.com/about/author-instructions/), 2) analysis packages and versions must be provided (e.g., what package or approach was used to generate the AUC; what parameters were selected, package versions, etc.), 3) an independent cohort must be used to validate the model, 4) all other issues identified by the reviewers including spelling errors, abbreviations, etc., and 5) expansion of the limitations to include additional factors like patient sex, comorbidities, analysis considerations, etc.

Reviewer 1 ·

Basic reporting

The researchers of this study developed a nomogram to predict the risk of chronic kidney damage
progression in IgAN patients. They analyzed clinical, pathological and laboratory data of a cohort of 443
Chinese biopsy-proven IgAN patients. The multivariate logistic regression analysis evidenced three
indicators as predictors of progression of the disease. They were body mass index, proteinuria and tubular atrophy/ interstitial fibrosis. The authors concluded that this nomogram may help the nephrologist to identify IgAN patients at high risk of progression of the disease.

Experimental design

The study is well done by the methodological point of view but there are some important limitations.
1. The paper cannot be accepted if the nomogram has not been validated in external independent cohort of IgAN patients. This point is very important because the number of IgAN patients included in this study to develop the nomogram is very low. Moreover, the patients were enrolled in only one Hospital.
2. The AUCs are 72 -74%. This means at limit of good prediction.

Validity of the findings

Major revision

Reviewer 2 ·

Basic reporting

No comment

Experimental design

No comment

Validity of the findings

no comment

Additional comments

In the manuscript, the authors enrolled 443 IgAN patinents and developed a nomogram model for predicting the risk of CKD progression. Since there have been many larger studies in the past and have constructed many risk models for predicting progression of IgAN, the lack of novelty is the shortcoming of the present study.
Here are some comments:
1.This is a single centre, small study with a relatively short follow-up time and the result of AUC was not ideal. Thus, my main concern is that a strong conclusion cannot be drawn from this study. I would suggest the authors to ‘soften’ the conclusion and discussions and to explain the statistical models in detail.
2.In the table1, why all quantitative data expressed as median ± interquartile range values, since normal distribution should be expressed as mean ± standard deviation values. Please explain.
3.Since the IgAN patients with different treatment methods, the severity and prognosis also vary. Thus, I would suggest performing subgroup analysis on patients treated with different methods, although the sample size may be relatively small.
4.Please explain all abbreviations at first use.
5.There are spelling errors in the text, please check and correct.

---

## Round 0.2 · Major Revisions

· Academic Editor

Major Revisions

Hello,

Thank you for your submission and revision. It does not appear that the code DOI was added to the manuscript. Please add and/or let us know where in the manuscript it was added.

Thank you.

Reviewer 1 ·

Basic reporting

Ok

Experimental design

Ok

Validity of the findings

OK

Additional comments

no comments

---

## Round 0.3 · accepted · Accept

· Academic Editor

Accept

Thank you for addressing the final issues. Congratulations--the manuscript is now ready for publication.